# Using a mobile nanopore sequencing lab for end-to-end genomic surveillance of *Plasmodium falciparum*: A feasibility study

**Aurel Holzschuh**[1,2☯]*, **Anita Lerch**[1☯], **Bakar S. Fakih**[2,3,4], **Safia Mohammed Aliy**[5], **Mohamed Haji Ali**[5], **Mohamed Ali Ali**[5], **Daniel J. Bruzzese**[1¤], **Joshua Yukich**[6], **Manuel W. Hetzel**[2], **Cristian Koepfli**[1]*

1 Department of Biological Sciences, Eck Institute for Global Health, University of Notre Dame, Notre Dame, Indiana, United States of America, 2 Swiss Tropical and Public Health Institute, Allschwil, Switzerland, 3 University of Basel, Basel, Switzerland, 4 Ifakara Health Institute, Dar es Salaam, United Republic of Tanzania, 5 Zanzibar Malaria Elimination Programme, Ministry of Health, Zanzibar, United Republic of Tanzania, 6 School of Public Health and Tropical Medicine, Tulane University, New Orleans, United States of America

☯ These authors contributed equally to this work.
¤ Current address: Department of Epidemiology and Microbial Diseases, Yale School of Public Health, New Haven, Connecticut, United States of America
* aholzsch@nd.edu (AH); ckoepfli@nd.edu (CK)

**Data Availability Statement:** All sequence data and code used in this analysis can be found at https://doi.org/10.5281/zenodo.10086238.

## Abstract

Genomic epidemiology holds promise for malaria control and elimination efforts, for example by informing on *Plasmodium falciparum* genetic diversity and prevalence of mutations conferring anti-malarial drug resistance. Limited sequencing infrastructure in many malaria-endemic areas prevents the rapid generation of genomic data. To address these issues, we developed and validated assays for *P. falciparum* nanopore sequencing in endemic sites using a mobile laboratory, targeting key antimalarial drug resistance markers and microhaplotypes. Using two multiplexed PCR reactions, we amplified six highly polymorphic microhaplotypes and ten drug resistance markers. We developed a bioinformatics workflow that allows genotyping of polyclonal malaria infections, including minority clones. We validated the panels on mock dried blood spot (DBS) and rapid diagnostic test (RDT) samples and archived DBS, demonstrating even, high read coverage across amplicons (range: 580x to 3,212x median coverage), high haplotype calling accuracy, and the ability to explore within-sample diversity of polyclonal infections. We field-tested the feasibility of rapid genotyping in Zanzibar in close collaboration with the local malaria elimination program using DBS and routinely collected RDTs as sample inputs. Our assay identified haplotypes known to confer resistance to known antimalarials in the *dhfr*, *dhps* and *mdr1* genes, but no evidence of artemisinin partial resistance. Most infections (60%) were polyclonal, with high microhaplotype diversity (median $H_E$ = 0.94). In conclusion, our assays generated actionable data within a few days, and we identified current challenges for implementing nanopore sequencing in endemic countries to accelerate malaria control and elimination.

**Funding:** This project was funded through internal funds of the University of Notre Dame made available to C.K. A.H. was supported by an Eck Institute for Global Health fellowship. The funders had no role in study design, data analysis, decision to publish, or preparation of the manuscript.

**Competing interests:** The authors declare that no competing interests exist.

## Introduction

Despite intensified control efforts, malaria remains one of the world's most prevalent and deadly infectious diseases, causing 247 million cases and more than 600,000 deaths in 2021, disproportionately affecting children under the age of five in sub-Saharan Africa [1]. Genomic epidemiology is a powerful approach to accompany malaria control and elimination efforts. Malaria parasite genetic data, incorporated into surveillance programs, can inform decision-making of malaria control programs by describing the *Plasmodium falciparum* population structure and relatedness to understand transmission dynamics [2–6], track the spread of anti-malarial resistance markers over space and time [2,7,8], or monitoring markers of diagnostic resistance (e.g., *hrp2/3* deletions) [9]. For example, there is increasing evidence of artemisinin partial resistance in several regions in Africa, including mainland Tanzania [10,11], Rwanda [12], and Uganda [13], threatening the effectiveness of artemisinin-based combination therapy (ACT) in Africa, which is the current front-line treatment for *P. falciparum* malaria. The World Health Organization (WHO) has recently emphasized the need to strengthen surveillance capacity by increasing technical and laboratory capacity and expanding data collection on antimalarial drug efficacy and resistance in Africa [14].

For genomic surveillance to inform malaria control programs in a timely manner, rapid generation of genetic data is needed [15,16]. Despite the potential benefits, many malaria endemic countries in Africa currently still have limited capacity for genomic sequencing, due to high costs, barriers to procurement, and an overall lack of sequencing and computing infrastructure [17]. Additionally, further capacity is needed to analyze and interpret generated genetic data. Nanopore sequencing, developed by Oxford Nanopore Technologies (ONT), offers a low-cost, portable, and rapid alternative to traditional sequencing methods in sites with minimal laboratory infrastructure, making it of particular interest for pathogen surveillance and clinical diagnostic applications [18,19]. Nanopore sequencing is suitable for whole genome sequencing (WGS), as well as targeted amplicon sequencing (AmpSeq), where specific loci of interest are targeted and sequenced after PCR amplification (i.e., amplicons). AmpSeq provides a high-throughput and, compared to WGS, more cost-effective and faster approach for genomic surveillance, including sensitive detection of *P. falciparum* minority clones in polyclonal infections, greater coverage at loci of interest, and less complex analysis pipelines [20]. Nanopore sequencing data can be generated and analyzed within hours to days of sample collection, allowing for rapid action. Previous studies have shown that nanopore sequencing can be used for genomic surveillance and diagnosis of multiple pathogens, including Ebola virus [21,22], Zika virus [23], Chikungunya virus [24], SARS-CoV-2 [25], and *Mycobacterium tuberculosis* [26,27]. Nanopore sequencing has previously been applied to *P. falciparum* for WGS [28] and AmpSeq of drug resistance genes [18,29].

Zanzibar, a semi-autonomous archipelago of the United Republic of Tanzania, has made considerable progress towards malaria elimination in recent years; however, progress has stagnated recently as new challenges arise in the last mile to achieving elimination [30,31]. At the current level of coverage, the combination of vector control, routine case management, and reactive case detection (RACD) implemented by the Zanzibar Malaria Elimination Programme (ZAMEP) is not expected to be sufficient to achieve elimination [32].

Importation of malaria from moderate or high transmission regions of mainland Tanzania to Zanzibar and the continuous receptivity of the environment on the islands is a main concern [6,33–35]. A recent genomic epidemiological study in Zanzibar found evidence of ongoing local transmission with spatial clustering of highly related parasites, and a substantial fraction of related parasite pairs between Zanzibar and mainland Tanzania [6]. Human movement has been associated with outbreaks and changing malaria transmission dynamics in

other sites [36]. Thus, understanding parasite importation is critical, and assessing its effect on transmission is necessary for targeting intervention packages that are tailored to the local transmission context [37,38]. Although Zanzibar has an efficient health surveillance system with good molecular laboratory infrastructure, current routine procedures are limited in their accuracy to distinguish local from imported infections, and to identify whether local foci are primarily due to imported infections or local transmission. Furthermore, the emergence of drug-resistant and diagnostic-resistant parasites (i.e., *hrp2/3* deletions) reported in East African countries including mainland Tanzania poses a threat to malaria elimination in Zanzibar [10–13,39,40]. Several markers of drug resistance have been previously reported in Zanzibar and thus warrant continued molecular surveillance [6,33].

Field-deployable nanopore sequencing assays for *P. falciparum* genetic diversity are not yet available, and assays for drug-resistance typing are scarce [18,29]. Here, we developed two multiplex AmpSeq panels, a 6-plex of highly polymorphic microhaplotypes, previously used in Zanzibar, and a 10-plex of known antimalarial drug resistance markers [6,41]. We have applied the method to mixtures of different strains of cultured parasites, mock DBS and RDT samples, and archived *P. falciparum* dried blood spot (DBS) field samples. Ultimately, we applied the method in a pilot study using a mobile laboratory setup in Zanzibar to evaluate the feasibility and applicability of in-country sequencing and to provide insight into the potential of nanopore sequencing as a tool for ongoing malaria surveillance in the context of malaria elimination at a representative site.

## Materials and methods

### Assay development and validation

**P. falciparum mock control samples and archived DBS for assay development and validation.** Mock control samples were made from several different *in vitro* cultured *P. falciparum* parasite strains (3D7/NF54, HB3, Dd2, FCB, V1/S, KH004, NHP1337, and NF54$^{C580Y}$) as described previously [42]. Briefly, *P. falciparum* parasite DNA was extracted using the Nucleo-Mag Blood 200 μL Kit (Macherey–Nagel) following the manufacturer's recommendations and quantified by highly sensitive and accurate droplet digital PCR (ddPCR) [43]. *P. falciparum* parasites were mixed at different proportions with uninfected human whole blood to obtain serial dilutions of different mixtures of control samples ranging from 25,000 to 2.5 parasites/μL blood. Mock control samples were stored at -20˚C until processing. We also included previously described [20,44] mock control mixtures of 3D7:HB3 with a relative abundance of 2% (98:2) and 1% (99:1) for the minority clone (HB3) at a density of 2,500 parasites/μL.

To evaluate the feasibility of nanopore sequencing from DBS and directly from rapid diagnostic tests (RDTs) and to compare the parasite detectability between the two sample types, we generated mock control DBS and RDT mixtures. Briefly, mock control DBS and RDTs were prepared by combining *in vitro* cultured *P. falciparum* parasites (NF54 and Dd2 at a 70:30 ratio) with uninfected human whole blood, at final parasite densities ranging from 25,000 parasites/μL to 2.5 parasites/μL. 50 μL of these mixtures were blotted onto Whatman 3MM Filter Paper (GE Healthcare Life Sciences), and 5 μL onto RDTs (SD BIOLINE Malaria Ag Pf) to mimic DBS and RDT samples. Mock DBS and RDTs were dried overnight and then stored in plastic bags with desiccants at -20˚C until processing.

Further, ten archived DBS samples from a previous field study conducted in Zanzibar between 2017 and 2018 [45] were included to compare performance of microhaplotypes between nanopore sequencing and the Illumina platform. Samples had previously been sequenced on an Illumina NextSeq 500 instrument and represented a range of parasite densities (range: 8–859,706 parasites/μL) at different multiplicity of infection (MOI) (range: 1–7) [6].

**DNA extraction of P. falciparum mock control samples.** For the mock control samples, DNA extraction was performed using the NucleoMag Blood 200 μL Kit (Macherey–Nagel) following the manufacturer's recommendations. DNA from mock DBS and RDT was extracted from both, entire 50 μL DBS and from entire RDT test strips, using a modified Tween-Chelex method as described previously [46], with minor modifications [42]; 400 μL of 10% Chelex 100 resin (catalogue #1422822, Bio-Rad Laboratories) in water was added instead of 150 μL to adjust for the higher DBS input volume. DNA was eluted in 250 μL nuclease-free water and stored at -20˚C until processing. Extracted DNA from mock control samples was quantified using ultra-sensitive *var*ATS qPCR as previously described [47].

**Multiplex PCR.** We developed two multiplex PCR panels, a 6-plex of highly polymorphic microhaplotypes to estimate multiplicity of infection (MOI) and genetic diversity, previously used in Zanzibar [6], and a 10-plex of known antimalarial drug resistance markers. The two panels can be used independently from each other, depending on the use case. All amplicons were amplified using previously published primer sequences [6,41,48], and no modifications were required for successful multiplexing (S1 **Table**). Mutations targeted by the drug-resistance panel are given in S2 **Table**. Median length across all amplicons is 231 bp (range: 179–250) for the microhaplotype panel, and 221 bp (range: 78–289) for the drug resistance panel. To create the two primer pools used in the multiplex PCRs, we combined 1 mM of each primer and diluted the combined primer mix to 5–15 μM per primer in nuclease-free water (NF dH2O). Each 25 μL multiplex PCR reaction consisted of 0.5 μL combined primer mix, 5 μL KAPA HiFi Buffer (5X), 0.75 μL KAPA dNTP Mix (10mM), 0.5 μL HiFi HotStart DNA Polymerase (1U/μL), 14.25 μL NF dH2O, and 4 μL sample template. PCR cycling conditions are provided in S3 **Table**. PCR reaction set-up and cycling conditions are identical for both multiplex PCRs.

After PCR, all samples including the positive and negative controls were inspected by gel electrophoresis to ensure successful PCR amplification (with blank negative controls) before proceeding to nanopore sequencing. 3 μL of either multiplex PCR was run for 30–45 minutes on a 2% agarose gel at 90V. Due to small size differences of the PCR products, individual amplicons cannot be distinguished by gel electrophoresis (only presence/absence of entire multiplex PCR). DNA was then quantified on a Qubit (ThermoFisher) fluorometer with Qubit dsDNA high sensitivity (Q32854) or Qubit dsDNA broad range kits (Q32853), as per manufacturer's instructions.

**Library preparation and nanopore sequencing.** The DNA library was prepared using the Ligation Sequencing Kit (SQK-LSK109; ONT) with Native Barcoding Expansion 1–12 and 13–24 (EXP-NBD104 and EXP-NBD114) or the Native Barcoding Kit 24 (SQK-NBD112.24; ONT) with Q20 chemistry according to the manufacturer's protocol (versions NBA_9093_v109_revN_12Nov2019 or NBE_9134_v112_revE_01Dec2021). The final sequencing library was quantified on a Qubit (ThermoFisher) fluorometer as described above, diluted in elution buffer (ONT) to a total of 50 fmol, and loaded onto R9.4.1 or R10.4 flow cells.

Sequencing runs used the MinION Mk1C sequencer (ONT) with MinKNOW software (distribution version 22.05.8, core version 5.1.0, and configuration version 5.1.5). All sequencing was performed using FLO-MIN106D or FLO-MIN112 R10.4 flow cells. Every run included 1 positive and 1 negative control.

Batches of 4–24 samples were sequenced in a total of 10 MinION runs. During the pilot study in Zanzibar, two batches of 24 samples were sequenced. Libraries were run between 2.5 h– 6.5 h (in 3 instances >20 h for flow cells with <350 pores left), aiming for approximately 5,000 reads per marker per sample. The time required for sequencing was variable, primarily dependent on the pores available on the flow cell. After stopping the run, flow cells were

washed with the Flow Cell Wash Kit (EXP-WSH004; ONT) as per manufacturer's instructions and reused (3–4 times) until the flow cell had <350 single pores available for sequencing at the start of the run.

**Data processing and bioinformatic methods.** *Basecalling.* Basecalling of raw nanopore reads was performed using Guppy (ONT) version 6.1.7 in GPU mode using the super-accurate (sup) model (dna_r9.4.1_450bps_sup.cfg or dna_r10.4_e8.1_sup.cfg). The following modifications to the standard settings were applied:

—device "cuda:0 cuda:1"–min_qscore 15 –do_read_splitting–max_read_split_depth 6 –trim_adapters–config dna_r9.4.1_450bps_sup.cfg (for ONT R9.4.1) or–config dna_r10.4_-e8.1_sup.cfg (for ONT R10.4). Basecalled read summary statistics were assessed with NanoStat using the–summary flag [49].

Due to the high-quality reads required by the DADA2 software, the unusual genome structure of *P. falciparum*, particularly the high A/T content (81%), and the expanses of repetitive, low-complexity regions, that are widely distributed throughout the genome [50], the minimum Q-score for passing reads was increased to a more stringent value of 15 to minimize erroneous reads.

Basecalling was not performed directly on the MinION Mk1C device but instead on a high-performance computing (HPC) cluster of the Center for Research Computing (CRC), University of Notre Dame, IN, USA. For the pilot study, raw sequencing data was transferred from ZAMEP to the HPC cluster but was slowed down by a slow internet connection.

Basecalled raw fastq sequence data that passed quality filtering criteria were then demultiplexed by sample native barcodes using guppy_barcoder (Guppy ONT version 6.1.7) with the commands–device "cuda:0 cuda:1"–trim_barcodes–min_score_barcode_front 75 –min_score_barcode_rear 75 –config configuration.cfg–barcode_kits "EXP-NBD104 EXP-NBD114" (for R9.4.1) or–barcode_kits "SQK-NBD112-24" (for R10.4).

*Haplotype calling.* After demultiplexing by sample, the demultiplexByMarkerMinION() function of R package HaplotypR version 0.5.0 (https://github.com/lerch-a/HaplotypR) [20] was used to demultiplex each sample by marker by identifying the forward and reverse primer sequences of each marker in all reads. The corresponding amplicon sequence within each primer pair was then extracted and saved to a new fastq file for each marker. Haplotype calling was done using the R package DADA2 (https://www.bioconductor.org/packages/release/bioc/html/dada2.html) [51]. Briefly, sequence reads were filtered to remove reads with ambiguous base calls (e.g., N) and incorrect sequence length. Haplotypes were called by using learnError() and dada() functions. Read counts per haplotype were then extracted with makeSequenceTable(). All haplotype calls were additionally filtered with removeBimeraDenovo() to remove chimeric haplotypes. Finally, to further remove false-positive haplotypes we defined and applied the following cut-off criteria: (1) each marker required a minimum of 100 total reads per sample, (2) haplotype calling required a minimum coverage of ≥65 reads per haplotype, and (3) a within-host haplotype frequency of ≥2%. The required minimum coverage of reads per haplotype was defined based on the number of reads stemming from barcode cross-talk in the sequencing run. Barcode cross-talk is a known challenge in the analysis of nanopore sequencing data referring to reads that are wrongly assigned during sample demultiplexing due to sequencing errors [52]. Barcode cross-talk was identified by counting the number of reads assigned to the wrong panel. The microhaplotype and drug resistance panels were amplified in separate sequencing runs using different barcodes, consequently, the drug resistance panel should not have any reads assigned to the microhaplotype panel and vice versa, allowing us to investigate barcode cross-talk. In 168 samples, with 16 markers each, resulting in a total of 2,688 markers sequenced, we observed 397 instances of reads stemming from barcode cross-talk. In total, 0.73% (21,072/2,872,368) of reads were assigned to the wrong barcode,

with an interquartile range (IQR) of 20–62 reads per cross-talk event. A cut-off of ≥65 reads per haplotype was chosen. The minority clone detection limit (i.e., within-sample haplotype frequency) was defined by the number of false-positive haplotypes found in all single-clone control samples. We observed 188 false-positive haplotype calls with an IQR frequency of 0.3–2.0%. The minority clone detection limit was set to ≥2% to minimize the amount of false-positive haplotype calls due to sequencing and amplification errors or other sources of noise.

### Pilot study

**Study site.** The Zanzibar archipelago in the United Republic of Tanzania has a population of approximately 1.8 million (2022 census) and consists of two main islands, Unguja and Pemba. The islands are divided administratively into 11 districts (7 in Unguja, 4 in Pemba). The study was conducted in two districts on Unguja, namely Mjini and Magharibi 'B', from September 28, 2022, to October 21, 2022 (**Fig 1**). The two districts were selected for two reasons: 1) The districts are closest to the ZAMEP laboratory (located in Magharibi 'B'), and 2) they reported the highest number of cases per year among all districts (S4 **Table**).

**Collection of field DBS and RDT samples from pilot study.** DBS were collected from individuals diagnosed with a *P. falciparum* monoinfection or mixed infection with *P. falciparum* and another species at health centers in the two study districts ("index cases"). Individuals were routinely tested by RDT (SD BIOLINE Malaria Ag *Pf* [histidine-rich protein II (HRP2)]/Pan [lactose dehydrogenase (pLHD)]). Index cases were followed-up routinely at their

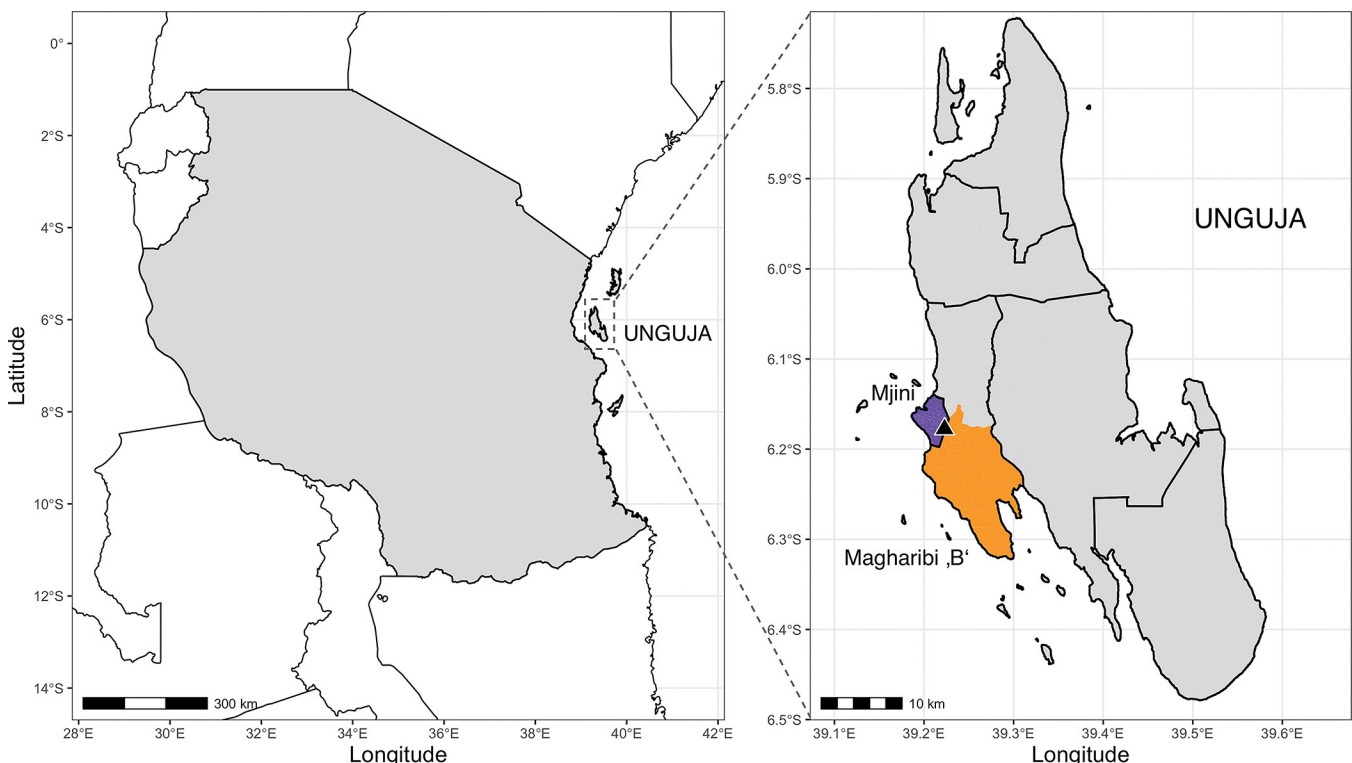

**Fig 1. Location of Zanzibar and study districts.** Left: Mainland Tanzania and location of the Zanzibar archipelago with the two main islands Unguja (in square) and Pemba. Right: Unguja with the two study districts Mjini (highlighted in purple) and Magharibi 'B' (highlighted in orange). The location of ZAMEP is indicated by black triangle. Administrative boarders used to generate the maps were downloaded from the Global Administrative Areas (GADM) database (available at: https://gadm.org/maps.html).

household by a District Malaria Surveillance Officer (DMSO) [53]. Household members of index cases one year and older were tested using RDTs ("follow-up" cases). From individuals that tested positive for *P. falciparum* by RDT (mono- or mixed infections), a DBS was collected from an additional finger-prick. Additionally, RDTs of *P. falciparum* positive individuals were collected whenever possible. An overview of sample collection is provided in S1 **Fig**.

**DNA extraction and quantification of field DBS and RDT samples from pilot study.** DNA was extracted from both, entire 50 μL DBS and from entire RDT test strips, using a modified Tween-Chelex method as described above for mock DBS and RDT samples. DNA was eluted in 250 μL nuclease-free water and stored at -20˚C until processing. Chelex-extracted DNA samples from DBS and RDTs were tested for *P. falciparum* using ultra-sensitive *var*ATS qPCR as previously described [47]. Note, qPCR quantification was not done as part of the pilot study in Zanzibar.

**Portable laboratory equipment and setup.** We used portable laboratory equipment (S2 **Fig**), Bento Lab and MinION, for development of assays in the lab at the University of Notre Dame and on-site genomic analysis of *P. falciparum*. The MinION Mk1C nanopore sequencer (ONT) is a portable sequencer with fully integrated computer and a screen. The Bento Lab (Bento Bioworks Ltd., United Kingdom) is a small mobile lab with dimensions of 330mm x 214mm x 81mm. It includes a 32-well PCR thermal cycler, microcentrifuge, and gel electrophoresis apparatus with LED transilluminator [54]. Further items for the portable laboratory equipment included the following: a Qubit (ThermoFisher) for library quantification, a heat block, a mini vortex, standard laboratory pipettes, and an external SSD drive (SanDisk Extreme Portable SSD, 1TB). The total amount of equipment could fit into one carry-on backpack or suitcase and weighs around 10 kg. Reagents for multiplex PCR and sequencing required frozen transport from the United States; this was achieved by use of packaging with cold packs in a Styrofoam box. Some reagents, including MinION flow cells require storage at 2–8˚C and were therefore transferred in a separate storage container with chilled cold packs. All laboratory procedures from DNA extraction to nanopore sequencing were performed in the ZAMEP laboratory in Zanzibar. A full list of laboratory equipment for the pilot study in Zanzibar is provided in S5 **Table** and an overview of the entire workflow is shown in **Fig 2**.

**Ethics statement.** Ethical approval for this study was obtained from the Zanzibar Health Research Ethics Committee (ZAHREC/04/PR/SEPT/2022/27) and the University of Notre Dame Institutional Review Board (approval no. 22-04-7215). Archived DBS used for methods validation were collected under approvals from the University of Notre Dame IRB (#18-12-5029), the Zanzibar Medical Research Ethics Committee (ZAMREC/0001/February/17), and the Ifakara Health Institute's Institutional Review Board (IHI/IRB/No: 003–2017). Informed written consent was obtained from all individuals before sample collection, or from their guardians in case of minors.

## Data analysis

**Multiplicity of infection and genetic diversity.** MOI was defined as the highest number of alleles detected by at least two of the 6 microhaplotype loci. The minimum of two loci was introduced to prevent overestimation due to false-positive allele calls. Population genetic diversity was estimated from the 6 microhaplotype loci by expected heterozygosity ($H_E$) using the formula, $H_E = \left[\frac{n}{n-1}\right]\left[1 - \sum p_i^2\right]$, where *n* is the number of genotyped samples and $p_i$ is the frequency of the $i^{th}$ allele in the population.

**Genetic relatedness.** Pairwise genetic relatedness was estimated using the R package *dcifer* version 1.2.0 (https://cran.r-project.org/web/packages/dcifer/index.html) an identity by descent (IBD)-based method to infer the degree of shared ancestry between polyclonal

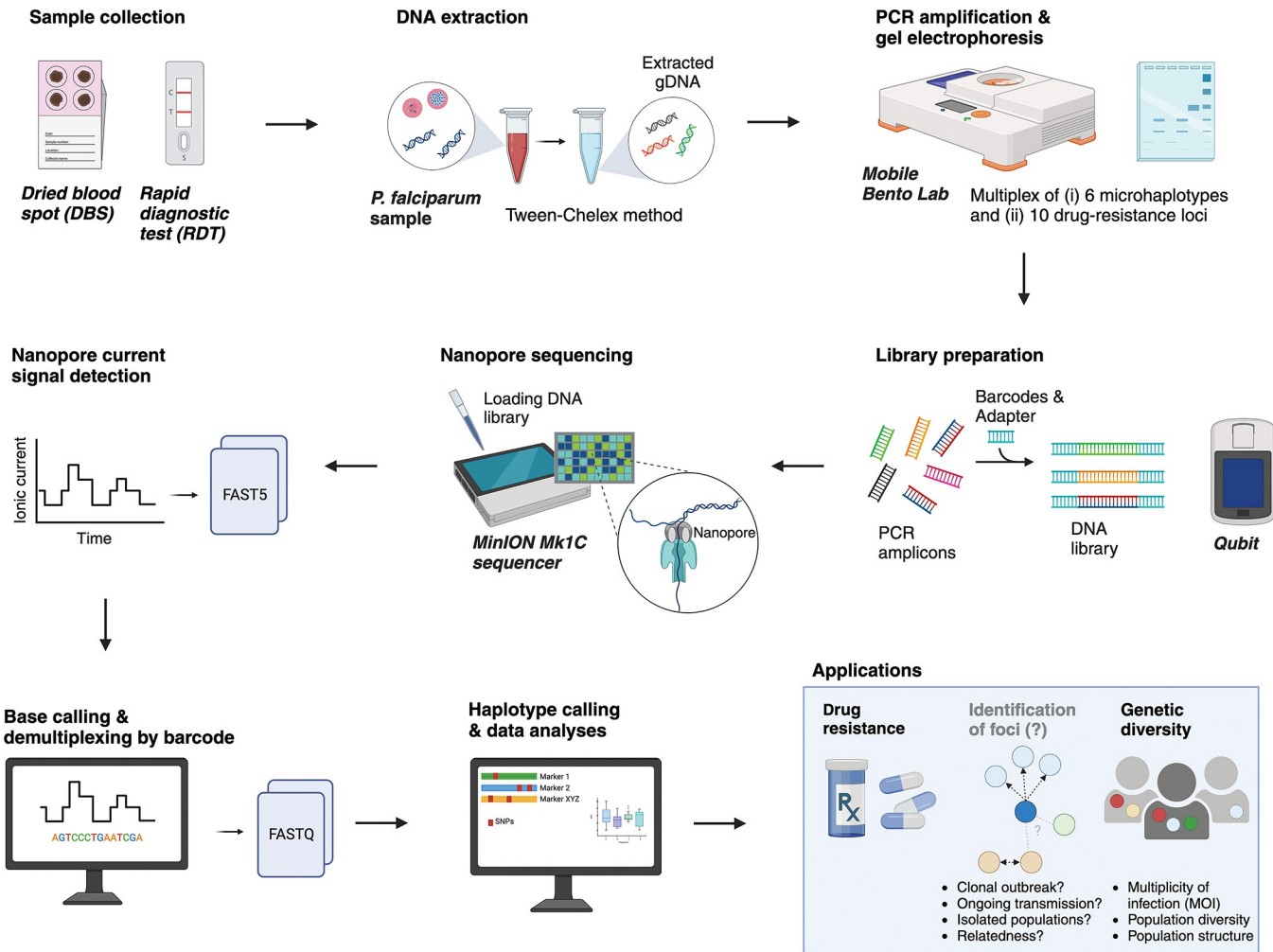

**Fig 2. Overview of the *P. falciparum* targeted nanopore sequencing approach using Oxford Nanopore Technologies (ONT) sequencing platform with the mobile Bento Lab.** Created with BioRender.com.

infections from unphased multiallelic data [55]. *Dcifer* provides likelihood-ratio *P*-values adjusted for one-sided tests. Briefly, from the samples, naïve MOI estimates and subsequently estimates of population allele frequencies adjusted for MOI were calculated. We used likelihood-ratio test statistic to test a null hypothesis that two samples are unrelated ($H_0$: IBD = 0) at significance level $\alpha$ = 0.05 (with the procedure adjusted for a one-sided test).

## Results

### Assay design and validation on mock control samples and archived DBS

We developed two multiplex PCR panels, a 6-plex of highly polymorphic microhaplotypes, and a 10-plex of known antimalarial drug resistance markers (S6 and S7 **Tables**). The workflow was validated on cultured *P. falciparum* controls, mock control DBS and RDT samples, and ten archived DBS field samples. We initially started with the R9.4.1 flow cell and then changed to R10.4 once available, which slightly increased the proportion of pass reads per run (i.e., ≥Q15; corresponding to ~97% accuracy) (S8 **Table**). Characteristics of all nanopore sequencing runs are shown in S8 **Table**. For both multiplex panels, reads across markers were

evenly distributed with high median read coverage across amplicon targets (range: 580x median coverage for *DR_dhps_436–437* to 3212x for *MH_cpmp_22*) (S3 **Fig**). Read numbers were higher for the microhaplotype panel than for the drug resistance panel.

Key antimalarial resistance markers genotyped in our assay matched the expected genotypes for all eight laboratory clones tested (**Table 1**). As observed previously [56], the Dd2 isolate contained both N86Y and N86F variants in *mdr1* due to having multiple copies of this gene. For lab mixtures with different number of clones and within-sample haplotype frequencies, the genotypes assigned matched the expected clones (**Table 2**). However, in Mixture 5 at a minority clone frequency of 5%, the S108T mutation fell below the defined cut-off, and would therefore be missed, and in Mixture 4 at a minority clone frequency of 2% two mutations (*dhps* A437G and *mdr1* N1042D) fell below the cut-off and two mutations (*dhfr* S108N and *mdr1* Y184F) were not detected (**Table 2**).

For the microhaplotype panel, all haplotype calls matched the expected haplotypes for all the four strains included (3D7/NF54, Dd2, FCB, HB3) when sequenced individually. In mixture samples, detection of individual clones was reflective of the clone frequencies, though there was some variation between different microhaplotypes (**Fig 3**A). Minority clones were detected at a within-sample haplotype frequency as low as 2% in almost all instances, indicating the suitability of the method for the detection of minority clones even at very low frequencies. Some alleles were missed because they fell below the applied cut-off, but respective reads were identified. Missed haplotypes could be explained by significantly lower number of reads in those samples, especially in more complex mixtures or at very low within-sample haplotype frequencies (**Fig 3**B). Of note is that for microhaplotype *MH_t73*, two false-positive alleles were called that were detected above the cut-off.

Further, we evaluated detectability of major and minor clones from mock DBS and RDT samples. The lowest parasite density for successful sequencing of both, microhaplotype and drug resistance panels, were 250 parasites/µL for mock DBS samples (S4A and S4B **Fig**) and 2,500 parasites/µL for mock RDT samples (S4C and S4D **Fig**), which was in line with visible PCR products on agarose gels. For most markers we were able to detect the major clone at those densities (**Fig 4**A). At 250 parasites/µL for DBS samples, the minority clone was detected at 8/11 dimorphic markers and at 7/11 dimorphic marker for the 2,500 parasites/µL RDT sample (**Fig 4**B). For five drug resistance loci, the two strains have identical alleles, therefore detection of minority clones cannot be shown. Detection of both the major and minor clones increased when no cut-off was applied for haplotype calling, even without bands visible on the gel (**Fig 4**C and 4D), indicating that in theory, detection of clones at very low densities would be possible. However, applying a cut-off is crucial to prevent false-positive allele calls, stemming from barcode cross-talk, amplification artifacts, or sequencing errors [20,52]. When the cut-off was applied, for both panels, no false haplotypes were identified in the DBS and RDT control mixtures.

Lastly, we compared the microhaplotype panel performance in terms of MOI estimates and concordance (Cohen's Kappa, κ) between nanopore and Illumina sequencing on ten archived field DBS samples from Zanzibar, that were Illumina-sequenced previously [6]. MOI estimates were nearly identical (**Table 3**), and agreement of haplotype calls was almost perfect, except for one microhaplotype that showed moderate agreement (**Table 4**).

## Sample characteristics pilot study

Overall, 20 *P. falciparum*-positive cases (9 males, 11 females) between the ages of 1 and 55 years were reported in the two study districts during the period of the pilot study. Most samples that were available for sequencing were from index cases (n = 15), with only two

**Table 1. Single-nucleotide polymorphisms (SNPs) and their corresponding amino acid mutations in the five drug resistance genes included in the multiplex panel as assessed by nanopore sequencing mock control samples of eight *P. falciparum* strains.** Parasite densities of all mixtures is 2,500 parasites/µL.

| | | Drug Resistance-Associated Gene and Change in Amino Acid[a] | | | | | | | | | | | | | | | | | | | | | |
| Strain and allele at each codon | | *dhfr* | | | | *dhps* | | | | | *mdr1* | | | | | | *mdr2* | | *k13* |
| Chromosome | | 4 | | | | 8 | | | | | 5 | | | | | | 14 | | 13 |
| Amino Acid Changes | | N51I | C59R | S108N/T | I164L | S436A/F | A437G | K540E | A581G | A613S/T | N86Y/F | Y184F | S1034C | N1042D | F1226Y | D1246Y | T484I | I492V | C580Y |
|---|---|---|---|---|---|---|---|---|---|---|---|---|---|---|---|---|---|---|---|
| 3D7/NF54[b] | Africa | N | C | S | I | S | G | K | A | A | N | Y | S | N | F | D | T | I | C |
| NF54^C580Y | | | | | | | | | | | | | | | | | | | Y |
| HB3 | C. America | N | C | N | I | S | A | K | A | A | N | F | S | D | F | D | T | I | C |
| V1/S | S.E. Asia | I | R | N | L | F | G | K | A | T | Y | Y | S | N | F | D | I | I | C |
| Dd2 | | I | R | N | I | F | G | K | A | S | F,Y | Y | S | N | F | D | I | I | C |
| FCB | | N | C | T | I | S | A | K | A | A | Y | Y | S | N | F | D | T | I | C |
| KH004 | | I | R | N | I | S | A | K | G | A | N | F | S | N | F | D | I | I | Y |
| NHP1337 | | I | R | N | L | S | A | K | G | A | N | Y | S | N | Y | D | I | I | Y |

[a] Gene IDs are *dhfr* (PF3D7_0417200), *dhps* (PF3D7_0810800), *mdr1* (PF3D7_0523000), *mdr2* (PF3D7_1447900), and *k13* (PF3D7_1343700). ALT residues to the 3D7 reference strain are shaded.

[b] 3D7 is a clone of NF54; as such, these are considered to be the same strain.

**Table 2. SNPs and their corresponding amino acid mutations in the five drug resistance genes included in the multiplex panel as assessed by nanopore sequencing of six *P. falciparum* control mixtures with different numbers of strains and at different clone frequencies.** Alleles that were below the defined cut-off setting and thus undetected, and alleles that were not detected at all (n.d.) are shaded. Parasite densities of all mixtures is 2,500 parasites/μL and corresponding absolute parasite density is given for all ratios.

Drug Resistance-Associated Gene and Change in Amino Acid

Gene / Chromosome: dhfr (4), dhps (8), mdr1 (5), mdr2 (14), k13 (13)

| Mixtures and allele at each codon | Strain [clone freq.] | Parasite density | N51I | C59R | S108N/T | I164L | S436A/F | A437G | K540E | A581G | A613S/T | N86Y/F | Y184F | S1034C | N1042D | F1226Y | D1246Y | T484I | I492V | C580Y | Total number of reads |
|---|---|---|---|---|---|---|---|---|---|---|---|---|---|---|---|---|---|---|---|---|---|
| **Mixture 1 [80:20]** | 3D7 [80] | 2,000 p/μL | N | C | S | I | S | G | K | A | A | N | Y | S | N | F | D | T | I | C | 18,764 |
|  | FCB [20] | 500 p/μL |  |  | T |  |  | A |  |  |  | Y |  |  |  |  |  |  |  |  |  |
| **Mixture 2 [70:30]** | 3D7 [70] | 1,750 p/μL | N | C | S | I | S | G | K | A | A | N | Y | S | N | F | D | T | I | C | 50,982 |
|  | Dd2 [30] | 750 p/μL | I | R | N |  | F |  |  |  | S | F,Y |  |  |  |  |  | I |  |  |  |
| **Mixture 3 [80:20]** | 3D7 [80] | 2,000 p/μL | N | C | S | I | S | G | K | A | A | N | Y | S | N | F | D | T | I | C | 20,614 |
|  | NF54^C580Y [20] | 500 p/μL |  |  |  |  |  |  |  |  |  |  |  |  |  |  |  |  |  | Y |  |
| **Mixture 4 [98:2]** | 3D7 [98] | 2,450 p/μL | N | C | S | I | S | G | K | A | A | N | Y | S | N | F | D | T | I | C | 13,084 |
|  | HB3 [2] | 50 p/μL |  |  | n.d. (shaded) |  |  | A (shaded) |  |  |  |  | n.d. (shaded) |  | D (shaded) |  |  |  |  |  |  |
| **Mixture 5 [35:5:60]** | 3D7 [35] | 875 p/μL | N | C | S | I | S | G | K | A | A | N | Y | S | N | F | D | T | I | C | 26,456 |
|  | FCB [5] | 125 p/μL |  |  | T (shaded) |  |  | A |  |  |  | Y | Y |  |  |  |  | T |  |  |  |
|  | HB3 [60] | 1,500 p/μL |  |  | N |  |  | A |  |  | A | N | F |  | D |  |  | T |  |  |  |
| **Mixture 6 [30:30:10:30]** | 3D7 [30] | 750 p/μL | N | C | S | I | S | G | K | A | A | N | Y | S | N | F | D | T | I | C | 14,425 |
|  | Dd2 [30] | 750 p/μL | I | R | N |  | F | G |  |  | S | F,Y | Y |  |  |  |  | I |  |  |  |
|  | FCB [10] | 250 p/μL |  | R | T |  |  | A |  |  | A | Y | Y |  |  |  |  | T |  |  |  |
|  | HB3 [30] | 750 p/μL |  | C | N |  |  | A |  |  | A | N | F |  | D |  |  | T |  |  |  |

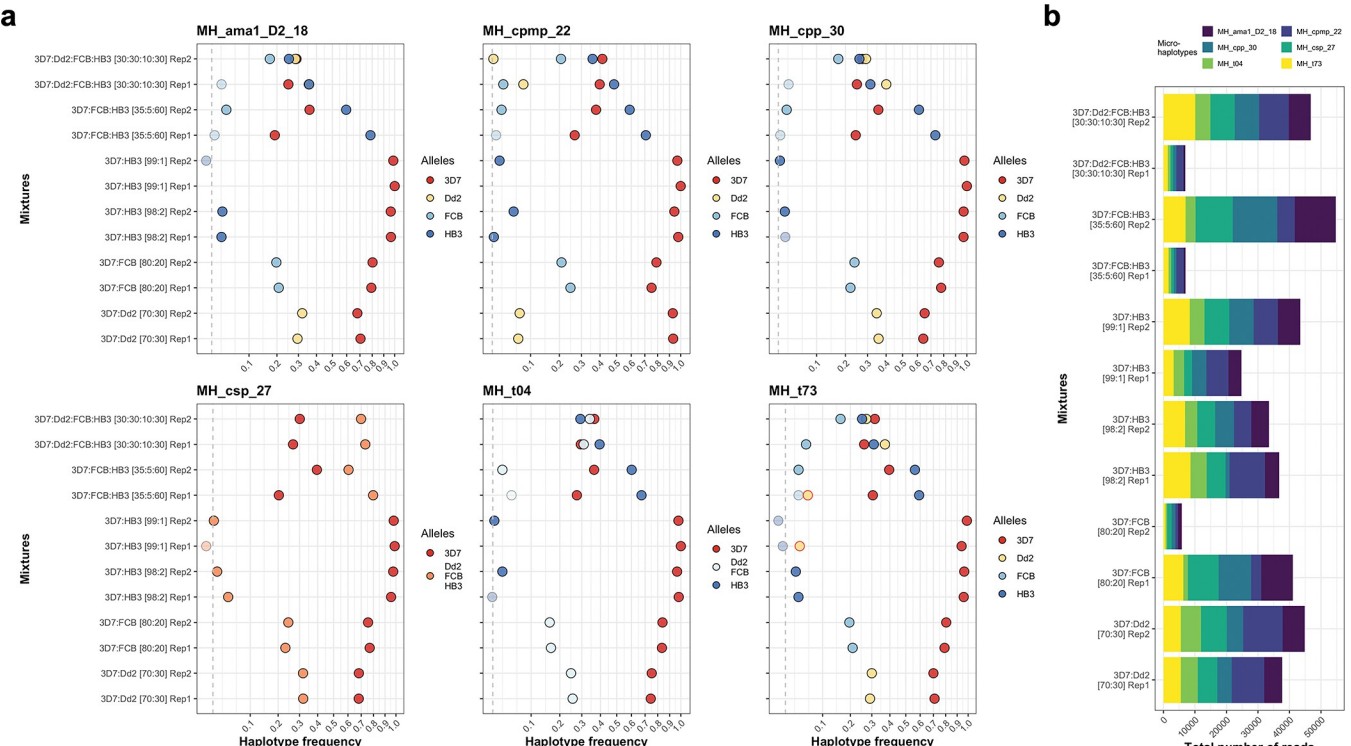

**Fig 3. Microhaplotype panel on different mixtures with different within-sample haplotype frequencies. A,** Haplotype calls for all six microhaplotype markers and their respective frequencies. Alleles that were detected but fell below the cut-off criteria (i.e., less than 65 reads per haplotype or within-host haplotype frequency below 2%) are shown in lighter color, false-positive haplotypes are indicated (red circle). Dashed grey line indicates the 2% within-sample haplotype frequency cut-off. Note that the x-axis is on square-root scale. **B,** Total number of reads for all mixtures. Reads per microhaplotype are indicated for each sample. Parasite density of all mixtures is 2,500 parasites /µL.

households harboring additional follow-up cases (n = 3 and n = 2, respectively). Recent travel to mainland Tanzania was reported by 50% of the individuals. DBS were collected from all 20 *P. falciparum*-positive individuals, and the original RDTs performed by the health worker or DMSO were obtained from 19 individuals. For DBS samples, 19/20 were positive by qPCR, and 18/19 samples were positive when extracted from RDTs (the negative DBS and RDT were not from the same individual). Infections were characterized by low parasite densities; mean parasite density in DBS samples after extraction was 808.5 parasites/µL (range: 0.2–10,173), and 354 parasites/µL (range: 0.1–5,478.5) for RDTs. Genome densities of extracted DNA are lower from RDTs as the volume of blood for DNA extraction is roughly 10-fold lower than from DBS (i.e., ~5 µL vs. ~50 µL).

## Nanopore sequencing of *P. falciparum* field samples from pilot study

All samples underwent both multiplex PCR amplification assays and were visually inspected by gel electrophoresis. Samples with observable bands in either of the two sample types by gel electrophoresis were taken forward for nanopore sequencing (11/20 DBS, 10/19 RDT; 10 paired samples). Subsequently, samples were quantified by qPCR at the University of Notre Dame. Mean parasite densities for samples that were sequenced were 1,468.5 parasites/µL (range: 25.9–10,173) for DBS and 707.7 parasites/µL (range: 1.9–5,478.5) for RDT samples, compared to 1.8 parasites/µL (range: 0.2–7.1) for DBS and 0.3 parasites/µL (range: 0.1–0.7) for RDT samples that were not sequenced (i.e., no visible bands). Due to logistical issues,

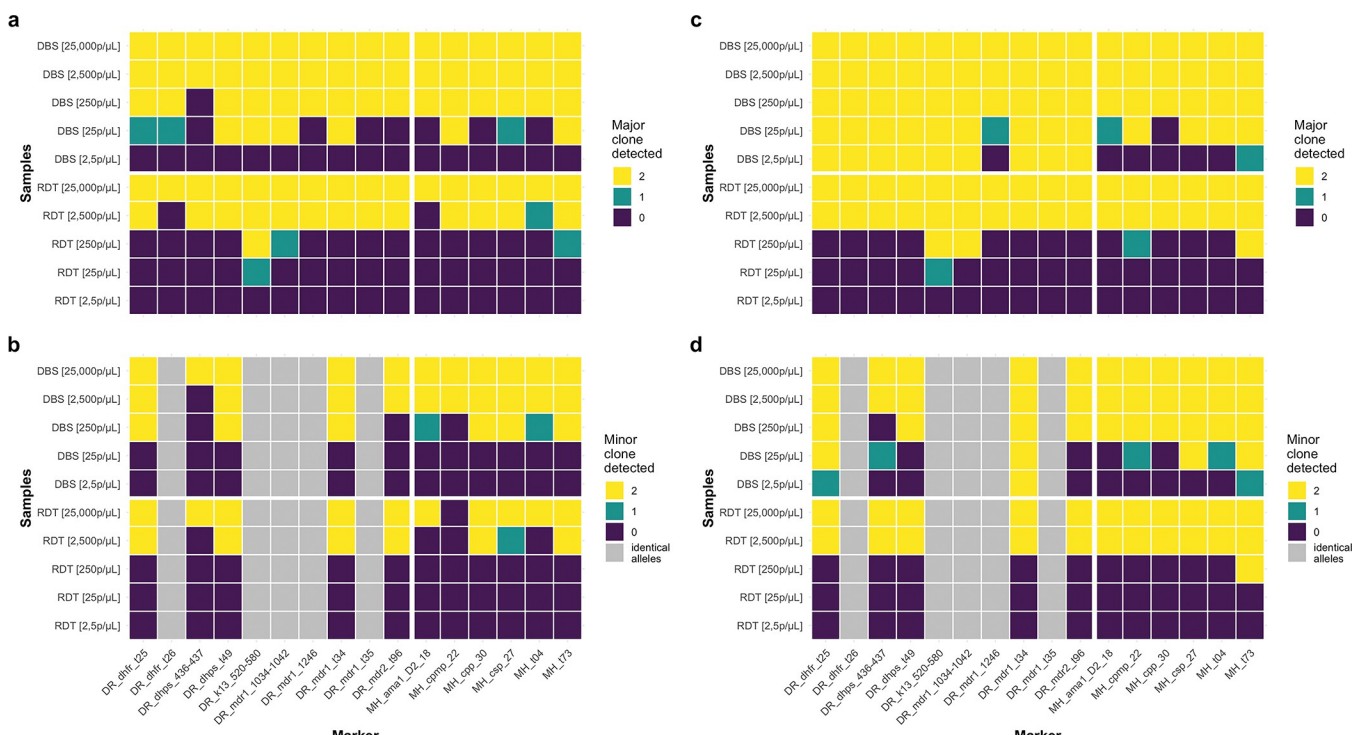

**Fig 4. Detectability of *P. falciparum* major and minor clones from mock DBS and RDT mixture samples. A**, Detection of the major clone at different parasite densities from DBS and RDT samples. **B**, Detection of the minor clone at different parasite densities from DBS and RDT samples. **C,** Same as **a** but without any cut-off applied. **D,** Same as **D** but without any cut-off applied. Colors indicate if the major clone was detected in both, one, or none of the replicates. All mixtures consist of NF54:Dd2 strains at a ratio of 70:30. Note that five drug resistance loci have identical alleles between the two *P. falciparum* strains, and thus minor clone detectability cannot be assessed (grey). Drug resistance loci are indicated by "DR" and microhaplotype loci by "MH" on the x-axis. Haplotypes were called with the defined cut-off criteria (see Methods).

microhaplotype marker *t04* could not be used in the pilot study in Zanzibar. Using ONT Q20 chemistry, two multiplexed batches of 24 samples each on the same R10.4 flow cell produced a mean of 16.8 Gb data, 1.21 M total reads, and 0.47 M pass reads called (≥Q15) per run (S8 **Table**). Median coverage across the markers ranged from 188x to 5253x (**Fig 5**), suggesting that low-volume blood samples from DBS and RDTs can be used as sample input.

**Table 3. MOI estimates from 10 archived field DBS samples, assessed using Illumina and nanopore sequencing.**
For both methods, the same cut-off criteria were used for haplotype calling (as defined in the Methods). MOI was estimated as the highest number of alleles detected by at least two of the 6 microhaplotype loci. Parasite density was determined by *var*ATS qPCR. For the two samples with the lowest density, reads for only one microhaplotype were obtained, preventing calculation of MOI.

| Parasites/μL | MOI$_{Illumina}$ (ref) | MOI$_{nanopore}$ | MOI difference |
|---|---|---|---|
| 859,706.2 | 2 | 2 | 0 |
| 310,130.7 | 2 | 2 | 0 |
| 31,678.7 | 7 | 7 | 0 |
| 3,369.5 | 6 | 7 | +1 |
| 835.6 | 3 | 3 | 0 |
| 222.1 | 2 | 2 | 0 |
| 69.7 | 2 | 3 | +1 |
| 27.1 | 2 | 1 | -1 |
| 18.9 | 1 | Only 1 microhaplotype | NA |
| 8.0 | 2 | Only 1 microhaplotype | NA |

**Table 4. Concordance of haplotype calls of 10 archived field DBS samples.** Haplotype calls of the 6 microhaplotypes obtained from nanopore and Illumina sequencing were compared with the same cut-off criteria applied.

| Microhaplotype marker | Cohen's Kappa (κ) | z-score | *P* value |
|---|---|---|---|
| MH_ama1_D2_18 | 0.921 | 12.7 | <0.001 |
| MH_cpmp_22 | 0.921 | 15.7 | <0.001 |
| MH_cpp_30 | 0.93 | 11.4 | <0.001 |
| MH_csp_27 | 0.956 | 13.2 | <0.001 |
| MH_t04 | 0.929 | 8.54 | <0.001 |
| MH_t73 | 0.78 | 14.9 | <0.001 |

## Drug resistance marker frequencies and parasite diversity of field samples

The frequency of parasites with molecular markers of drug resistance were comparable between DBS and RDT samples (**Fig 6**A), and reflective of previously reported drug resistance frequencies in the examined genes [6,33]. Some mutations in the *dhfr* and *dhps* genes have reached fixation. No *k13* mutations associated with artemisinin partial resistance were identified. Most infections were polyclonal, from both DBS (63.6%) and RDT (60%) samples, as reported previously in Zanzibar [6]. Median MOI was 2 (IQR 1–4) for DBS samples and 2 (IQR 1–2) for RDT samples (**Fig 6**B). Genetic diversity for the five microhaplotypes used was high and comparable for DBS (mean $H_E$: 0.94 ± 0.02) and RDTs (mean $H_E$: 0.93 ± 0.01) (**Fig 6**C).

Of the 10 paired DBS and RDT samples, eight were identical (i.e., identity-by-descent (IBD) = 1), one with an IBD estimate of 0.67 (Sample 15), and one was completely unrelated (IBD = 0; Sample 14). RDT and DBS for Sample 14 were collected four days apart, thus

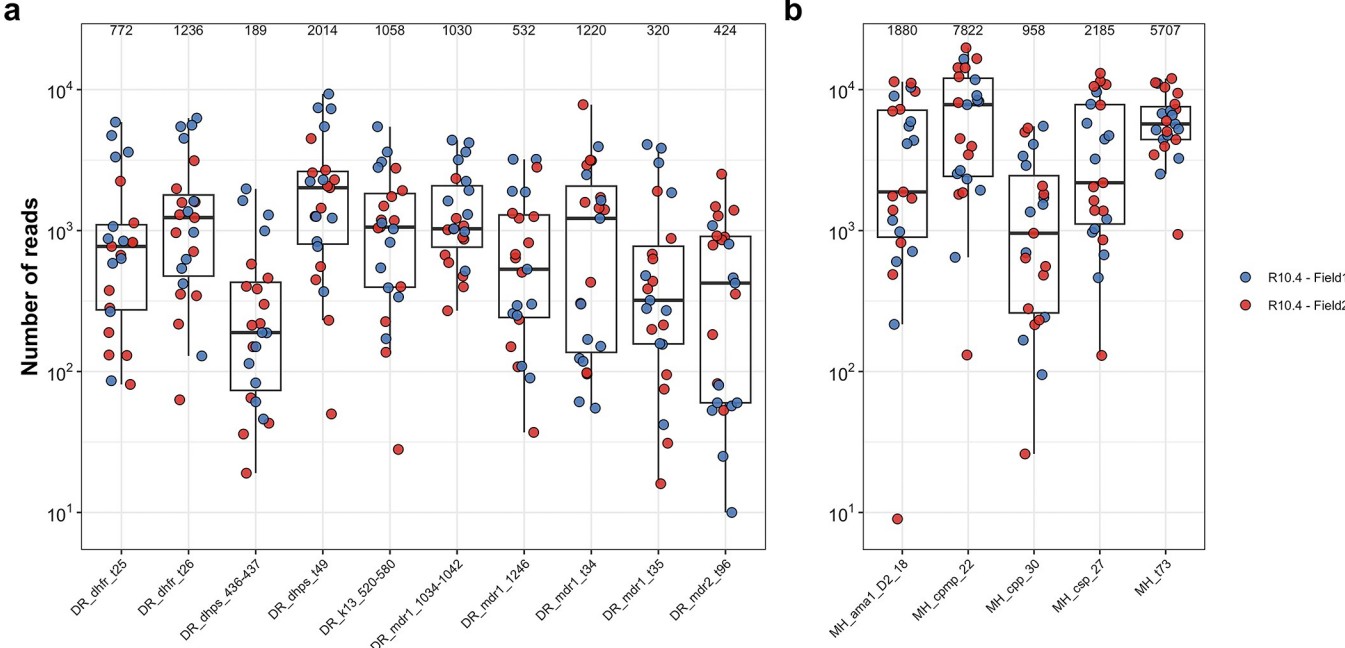

**Fig 5. Coverage per marker for (a) drug resistance panel and (b) microhaplotype panel.** Y-axis shows number of reads (log10) covering each amplicon target per sample for each of the two MinION runs for field samples from the pilot study in Zanzibar. Median coverage for all amplicons is indicated on top of the graph. Positive controls were excluded. The box bounds the IQR divided by the median, and Tukey-style whiskers extend to a maximum of 1.5 × IQR beyond the box.

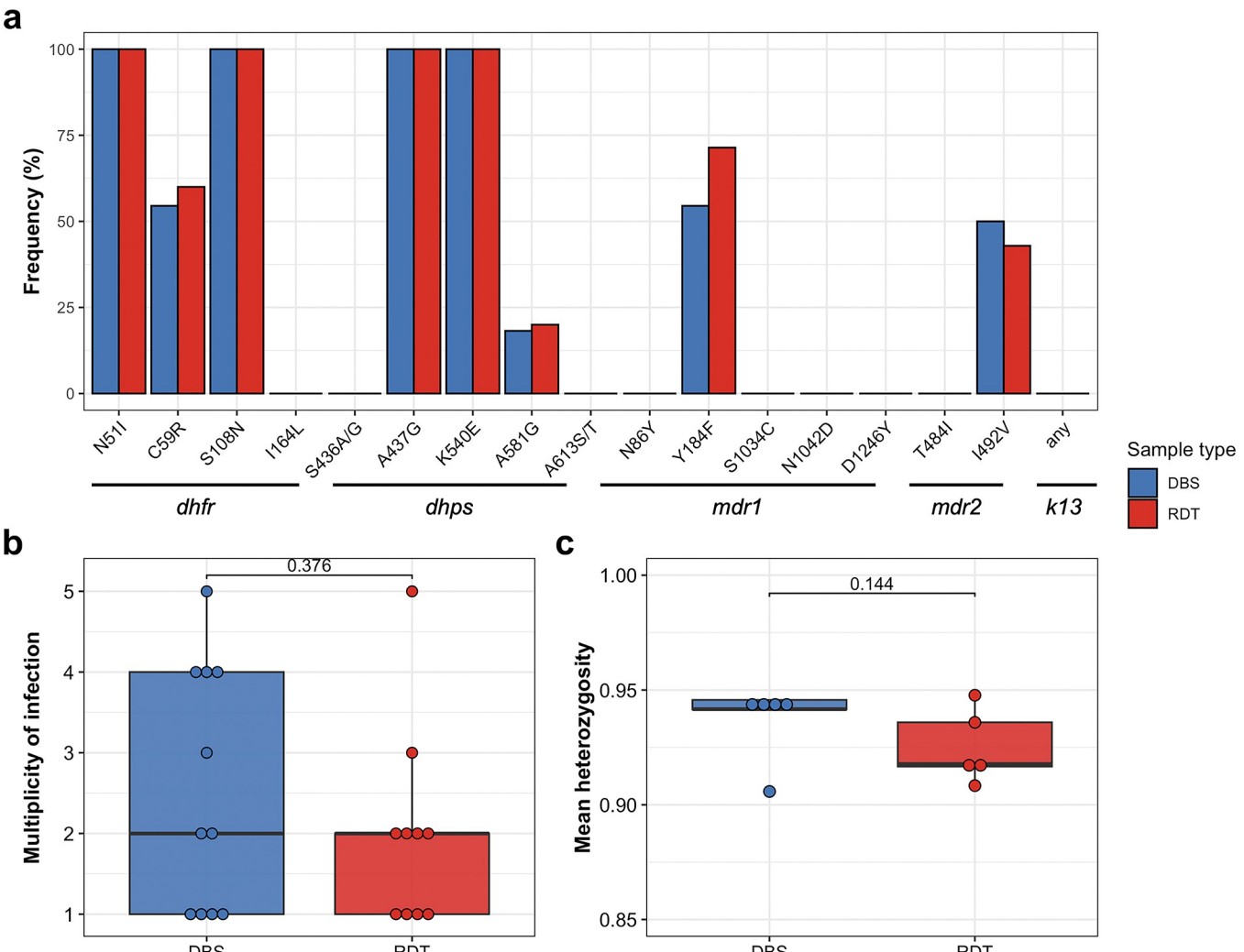

**Fig 6. Drug resistance frequencies and genetic diversity of *P. falciparum* samples from the pilot study in Zanzibar. A**, Frequency of mutations in the five drug resistance genes covered by the drug resistance panel for DBS (n = 11) and RDT (n = 10) samples collected during the pilot study in Zanzibar. Note that in the *dhps* mutation **G**437A the reference is resistant, hence this is re-coded as A437**G** and values indicate the frequency of the reference allele. **B**, Multiplicity of infection and **C**, heterozygosity of the same samples using the microhaplotype panel. The box bounds the IQR divided by the median, and Tukey-style whiskers extend to a maximum of 1.5 × IQR beyond the box.

antimalarial treatment likely influencing genotyping results. When samples from different individuals were compared, two significantly related pairs were identified (IBD: 1 and 0.68, respectively) within the DBS samples (ID 1–6; ID 10–20) and two (IBD: both 1) within the RDT samples (ID 1–6; ID 10–14) (**Fig 7**). Sample pair 10–14 had IBD = 1 for both, RDT and DBS.

## Current challenges to implement in-country nanopore sequencing in Zanzibar

During the pilot study in Zanzibar, we aimed to assess the feasibility of in-country nanopore sequencing in a representative site. Several current challenges identified are presented in **Table 5**.

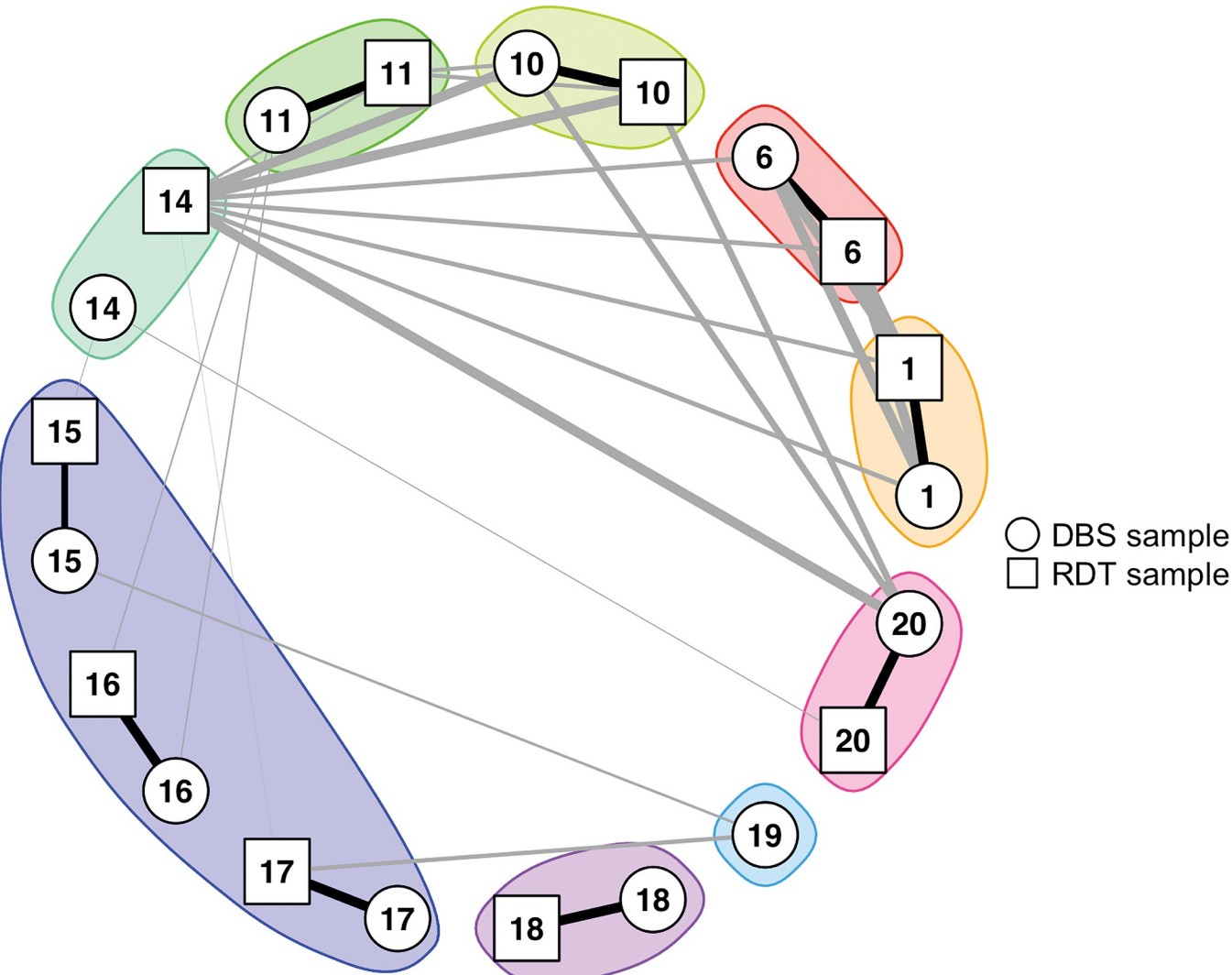

**Fig 7. Relatedness network by identity-by-descent (IBD) for pairs of isolates from 11 individuals from Zanzibar.** Nodes with identical IDs are from the same individual, either collected as DBS (circle) or as RDT (square) sample. Each RACD investigation is highlighted by a different color. Edges between the same individual (but different sample type) are in black and between other isolates are in grey. Edge thickness corresponds to IBD values. All IBD estimates are plotted, irrespective of statistical significance (e.g., for sample pair 14–10, IBD = 0.99, but *p* = 0.14). Note, the dark blue cluster is the only RACD investigation with an index case (ID: 15) and two follow-up cases (ID: 16 and 17). IDs are in chronological order.

## Discussion

Genomic surveillance can support malaria control and elimination efforts, for example by improving our understanding of malaria transmission, monitoring drug resistance markers, or aiding vaccine development [16,41,48,57,58]. We developed and validated two multiplexed nanopore AmpSeq panels targeting *P. falciparum* microhaplotypes and markers of drug resistance from DBS and RDT samples. We successfully sequenced clinical and asymptomatic infections and performed genomic analysis of *P. falciparum* in a pilot study in Zanzibar using a mobile laboratory. We observed high parasite genetic diversity and MOI, and high frequencies of mutations in the *dhfr*, *dhps*, *mdr1*, and *mdr2* genes associated with drug resistance to sulfadoxine-pyrimethamine and lumefantrine, but no *k13* mutations. These findings corroborate results from previous studies using Illumina sequencing [6,33]. We successfully sequenced

**Table 5. Current challenges to implement in-country nanopore sequencing in Zanzibar.**

| Challenges | Possible solutions |
|---|---|
| **1. Sample collection**<br>▪ **Collection of DBS before treatment of individuals.** | ▪ Well-organized RACD system already in place.<br>▪ Include DBS collection in routine-surveillance procedure (collect DBS together with RDT)<br>▪ Store routinely collected RDTs if available. |
| **2. Laboratory work and capacity**<br>▪ **Basic understanding of nanopore sequencing needed.**<br>▪ **Library preparation and sequencing is complex.**<br>▪ **Experienced lab person desirable (e.g., for troubleshooting).**<br>▪ **Basic sequencing infrastructure needed.** | ▪ Laboratory training of several months to improve practical and theoretical understanding of nanopore sequencing and general good laboratory practice.<br>▪ Laboratory supervisor or online support (e.g., via Zoom) if not available.<br>▪ Feasibility of portable laboratory around MinION (ONT) demonstrated. |
| **3. Procurement of sequencing consumables and reagents**<br>▪ **Sequencing devices, consumables and reagents are challenging to procure and expensive.** | ▪ Short-term: Gaps in procurement can be filled in part by collaborating Northern institutions.<br>▪ Long-term: Robust supply chains and investment from the major suppliers of sequencing technology are needed, without reliance on procurement abroad. |
| **4. Bioinformatics and computational capacity**<br>▪ **Basic bioinformatic skills needed (e.g., command-line interface, R programming language) as informatics workflow is advanced**<br>▪ **Access to HPC clusters for 'sup' basecalling needed.**<br>▪ **Access to high bandwidth internet connectivity desirable to transfer raw sequencing data to HPC clusters.** | ▪ Short-term: Outsourcing bioinformatics to partner institutions (with fast turn-around time).<br>▪ Long-term: Build up a cohort of bioinformaticians, develop local expertise and bioinformatics systems capacity (e.g., shared HPC clusters or high-performance desktop computer with GPU capability). |
| **5. Data analysis**<br>▪ **Basic understanding of molecular epidemiology, parasite genetics (e.g., molecular metrics such as MOI), and population genetics needed.**<br>▪ **Disease and public health specialists need to be able to interpret and understand the generated data for policy and decision-making.** | ▪ Short-term: Outsourcing of data analysis to partner institutions.<br>▪ Long-term: Training personnel such as bioinformaticians, molecular epidemiologists, and public health specialists. |
| **6. Data storage**<br>▪ **Adequate long-term storage needed.** | ▪ Cloud storage systems provide low-cost storage (but privacy concerns must be addressed); or alternatively hardware for long-term storage is available. |

infections extracted from RDTs despite the small volume of blood put on RDTs (~5 µL). RDTs are widely used in routine diagnosis of suspected malaria patients attending health facilities in malaria-endemic countries and during RACD by ZAMEP [45,53,59] and other malaria control and elimination programs. The possibility to use routinely collected positive RDTs can facilitate large-scale genomic surveillance as no additional blood samples need to be collected and no PCR screening to identify positive samples is required.

## Assay development and validation on mock control samples and archived DBS

Both, the drug-resistance and microhaplotype panels, showed relatively even amplification of all markers included in each multiplex reaction. Both panels were able to identify the major clone down to densities of 250 parasites/µL for DBS samples and 2,500 parasites/µL for RDT samples. Of note, approximately half of DBS and RDTs collected in Zanzibar were not sequenced as they did not have observable bands on the gel. All of these were of low parasite density as confirmed by qPCR. Our informatics workflow was designed to capture all information from polyclonal infections (i.e., utilizing all the alleles detected in a polyclonal infection) by adapting existing short-read haplotype inference tools, HaplotypR [20] and DADA2 [51].

This is relevant as strains of interest (e.g., carrying drug resistance mutations) might be present as minority clones. When compared to Illumina sequencing results, concordance of haplotype calls was almost perfect and MOI estimates nearly identical. Minority clones were detected at a very low within-sample haplotype frequency of 2% in almost all instances. However, our analysis workflow is rather complex, requiring knowledge of R programming language and access to HPC clusters. Such services could be purchased from platforms like Google Cloud, Microsoft Azure, or AWS, but will require knowledge of cloud computing and funds to cover them on an as-needed basis. The future goal should be to implement the workflow into an easy-to-use graphical user interface tool to allow rapid analysis and visual interpretation of parasite populations or drug resistance mutations.

Setting a threshold for the inclusion of minority clones is a trade-off between improved detection of minority clones and false-positive calls. False-negative haplotypes were missed in most cases when they fell below the cut-off threshold, usually at low read numbers. False-positive allele calls were rare. With no cut-off applied, the detection of both the major and minor clones increased dramatically, especially in DBS samples, but resulted in a higher number of false-positive allele calls. With the development of new flow cells and kits that exhibit higher accuracy, cut-off criteria could be lowered, potentially resulting in more sensitive sequencing of low-density samples. Upgrading the flow cells and chemistries from R9.4.1 to R10.4 (with Q20 chemistry) slightly improved the proportion of reads passing the minimum Q-score threshold ($\geq$15). The latest available nanopore chemistry Q20+ (kit 14, R10.4.1 flow cells), is promising improved flow cell performance with greater raw read accuracy and increased throughput [60]. In addition, ONT's new basecalling software *Dorado* (https://github.com/nanoporetech/dorado/) is expected to further improve raw read accuracy and overall performance.

## Feasibility of nanopore sequencing in Zanzibar

We performed multiplexed nanopore sequencing with the entire workflow in-country at the ZAMEP laboratory, including sample collection embedded in routine surveillance-response activities, DNA extraction, PCR, library preparation, sequencing, and initial data analysis. End-to-end time from sample collection to completed sequencing for a multiplexed batch of 24 samples was around 3 days. Basecalling and haplotype calling were done on site and took an additional day, but was impeded by poor bandwidth internet connectivity to transfer data to HPC clusters. Real-time basecalling directly on the MinION Mk1C device is available, but this precludes the most accurate basecalling model (i.e., sup), resulting in lower read quality. Investing in a powerful computer would be an appropriate option, allowing the use of the most accurate basecalling models. While a commercial gaming laptop may offer some convenience and would be comparatively inexpensive, it has storage and GPU limitations. Setting up a high-performance desktop computer with GPUs, costing around US$3,000 to US$8,000, and using hard drives to transfer raw sequencing data would be the best option. This investment would enhance the mobile nanopore sequencing lab, saving time and reducing dependence on internet connectivity and HPC clusters.

A direct comparison of haplotypes between DBS and RDTs was only possible to a limited extent, and discrepancies between the two sample types were expected because DBS and RDTs were collected 1–4 days apart (S1 **Fig**) and the blood volume for RDT samples (~5 μL) is roughly 10-fold lower than for DBS samples (~50 μL), hence reducing sensitivity to detect minority clones. Additionally, all index cases (majority of samples) were treated at initial diagnosis, hence the parasites detected in the DBS sample at follow-up are not necessarily exactly reflective of the original RDT sample. The success of sequencing low-density parasite samples

depends on the amount of template DNA available. Ways to potentially increase the number of samples successfully sequenced include, but are not limited to, DNA enrichment by selective whole genome amplification, increasing the volume of blood collected (i.e., collect as whole blood stored in EDTA tubes), using a more sensitive DNA extraction method, concentrating the DNA (e.g., elute in smaller volume or use SpeedVac), or increasing PCR cycles.

For genomic epidemiology to be effective in clinical and public health settings, a quick turn-around time is crucial [16,21,22]. For example, drug resistance is often initially discovered through patients presenting with recurrent parasitemia within a few weeks of treatment, or through observation of delayed clearance times. In cases of recurrent parasitemia, the microhaplotype panel allows to distinguish between reinfection with a different clone, or recrudescence, i.e., true treatment failure. In case of flare-ups of case numbers in a low transmission setting, the microhaplotype panel can inform on whether the increase is a result of local transmission, e.g., identifying closely related infections. This observation can then support the identification of suitable targeted interventions to address local transmission, e.g., through targeted vector control. Alternatively, increased case numbers might be the result of (an increase in) importation of infections, potentially followed by short local transmission chains. This information might lead to increased efforts to reduce importation of cases. The drug resistance panel could be useful for periodic country-wide monitoring of the frequency of drug resistance markers. This may provide an early signal of increasing drug resistance, identify subnational foci of emerging drug resistance, and the need to assess the efficacy of a particular drug through appropriate therapeutic efficacy studies.

Conducting sequencing in-country, rather than transporting samples across borders, can significantly reduce time to results. To promote decentralization of genomics and ensure more equitable availability of genomics capacity, it is necessary to establish sequencing workflows in endemic country settings. Investing in sequencing capacity includes the strengthening of technical skills and establishment of technology that can also be applied to pathogens and emerging infectious diseases other than just malaria [61]. A close coordination and collaboration between different disease control programs is therefore important. Upfront hardware expenses are modest (e.g., US$4,900 for MinION Mk1C including 6 flow cells, US$2,000 for Bento Lab, and US$4,000 Qubit). Additional equipment needed, e.g., pipettes, heat blocks, are often available in molecular surveillance labs, and else cost is less than US$2,000. Running costs are estimated to be around US$25.00 per sample (per panel) using this workflow with multiplexed batches of 24 samples and reusing flow cells 2–3 times. ONTs washing procedure for reusing flow cells has been shown to be very effective with less than 0.1% of any previously loaded sample contaminating a subsequent run [62]. The main consumable cost is the flow cell at approximately $900, thus costs per sample are heavily impacted by the number of samples run in parallel, and the number of times the flow cell is reused. However, by ordering flow cells in bulk, costs can be significantly reduced. Estimating the cost of genomic data processing is difficult, however, computing costs to process genomic data using AWS cloud service with optimal hardware configurations was recently estimated to be as low as $0.29 per *P. falciparum* genome, a 4.6-fold reduction in cost compared to standard pipelines [63].

This study also identified key challenges beyond infrastructure in implementing in-country nanopore AmpSeq in the context of Zanzibar (**Table 5**). These challenges must be addressed in the frame of establishing a robust in-country malaria sequencing program in Zanzibar and other African countries. Furthermore, these challenges are likely to apply to control and elimination programs targeting not only malaria but other diseases (e.g., soil-transmitted helminths) as part of broader pathogen genomics initiatives [17,61]. For an efficient integration of molecular surveillance into routine surveillance-response activities at ZAMEP, priority should be given to investing in strengthening the limited availability of reagents and

consumables, shortage of technical capacity (particularly in the areas of laboratory work), bio-informatics challenges, data sharing and privacy concerns, power and internet access limitations, and quality control standards (such as SOPs, supervisors, etc.).

In conclusion, we developed and field-tested protocols and an analytical workflow for nanopore AmpSeq, highlighting its benefits for antimalarial drug resistance surveillance and characterizing within-host and population-level diversity. Nanopore sequencing platforms in combination with a mobile laboratory such as the Bento Lab offer a promising solution for improving research equality in countries fighting malaria. These portable platforms enable sequencing in-country, empowering local researchers and national control programs to make informed programmatic decisions. They are a promising tool for genomic analysis and surveillance in research centers with limited facilities in the frame of comprehensive efforts of scientific and programmatic capacity strengthening.

## Supporting information

**S1 Checklist. Inclusivity in global research checklist.**
(DOCX)

**S1 Fig. Flow chart of field sample collection.** Illustrated from initial diagnosis to reactive response at a malaria patient's household. Collection of RDT and DBS samples from both, "index" cases and "follow-up" cases are indicated. A total of 20/20 DBS and 19/20 RDTs were available. DBS = dried blood spot; HF = health facility; HH = household; RDT = rapid diagnostic test.
(TIFF)

**S2 Fig. Key equipment of the mobile nanopore sequencing laboratory setup at the ZAMEP laboratory in Zanzibar.** (**a**) Bento Lab (Bento Bioworks Ltd.) that includes a 32-well PCR thermal cycler, microcentrifuge, and gel electrophoresis apparatus with LED transilluminator. (**b**) minION Mk1C sequencer (ONT) with fully integrated computer and a screen. (**c**) Qubit (ThermoFisher) that is needed for DNA quantification during sequencing library preparation. (**d**) standard laboratory pipettes. Other equipment (not shown here) included a heat block, a mini vortex, and an external SSD drive (SanDisk Extreme Portable SSD, 1TB). For a detailed list see S5 Table.
(TIFF)

**S3 Fig.** Coverage per marker in control samples for (**a**) drug resistance panel and (**b**) microhaplotype panel. y-axis shows number of reads (log10) covering each amplicon target per sample for each of the MinION runs, including control samples from the two MinION runs in the field in Zanzibar. Median coverage for all amplicons is indicated on top of the graph. Note, archived field DBS samples and low-density DBS and RDT control samples (<250 parasites/μL) were excluded.
(TIFF)

**S4 Fig. Gel electrophoresis of multiplex PCR products of the microhaplotype and drug resistance panels from mock control DBS and RDT NF54:Dd2 mixture at 70:30 ratio and at different parasite densities.** DNA was extracted from 50 μL DBS, or from RDTs with 5 μL blood spotted. **a**, Microhaplotypes from DBS. **b**, Drug resistance markers from DBS. **c**, Microhaplotypes from RDTs. **d**, Drug resistance markers from RDTs. Mock mixtures were spotted on DBS or RDTs at different parasite densities ranging from 25,000 parasites/μL– 2.5 parasites/μL. For DBS, 50 μL was blotted onto filter papers to mimic DBS samples. For RDTs, 5 μL were blotted onto RDTs to mimic RDT samples. All samples were extracted and assessed in

duplicate. For all samples, 3 μL were loaded onto a 2.5% agarose gel and run at 90V for 1 hour. DBS = Dried Blood Spots. RDT = Rapid Diagnostic Test. Neg1 = Negative control (only human whole blood used as template). Neg2 = Negative control (nuclease free water used as template).
(TIFF)

**S1 Table. Primer sequences for the 6-plex microhaplotype panel and the 10-plex drug resistance panel.** Appropriate volume of stock concentration per primer was diluted in NF dH2O (added up to 1mL) to get concentration of final pool. Pool concentrations and final concentrations of each primer are indicated.
(DOCX)

**S2 Table. Antimalarial drug resistance markers and nucleotide positions of known mutations.**
(DOCX)

**S3 Table. Thermal cycling conditions of 6-plex microhaplotype panel and 10-plex drug resistance panel.**
(DOCX)

**S4 Table. Annual number of malaria cases reported through the individual case-based Malaria Case Notification system, by island and year.** Magharibi 'B' and Mjini are highlighted in orange and purple, respectively. Source: ZAMEP.
(DOCX)

**S5 Table. Full list of laboratory equipment for the pilot study in Zanzibar.**
(DOCX)

**S6 Table. *P. falciparum* antimalarial drug resistance genes targeted by the 10-plex amplicon panel.**
(DOCX)

**S7 Table. *P. falciparum* highly polymorphic microhaplotypes targeted by the 6-plex amplicon panel.**
(DOCX)

**S8 Table. Individual nanopore sequencing run characteristics.**
(DOCX)

## Acknowledgments

We would like to thank all patients and household members of index cases who participated in our study, the DMSOs in the study districts who were responsible for sample collection, and the colleagues at ZAMEP for their support. We thank Dr. Michael T. Ferdig, Lisa A. Checkley and John Kane from the University of Notre Dame for providing cultured *P. falciparum* parasite strains.

## Author Contributions

**Conceptualization:** Aurel Holzschuh, Joshua Yukich, Manuel W. Hetzel, Cristian Koepfli.

**Data curation:** Aurel Holzschuh.

**Formal analysis:** Aurel Holzschuh, Anita Lerch.

**Funding acquisition:** Cristian Koepfli.

**Investigation:** Aurel Holzschuh, Anita Lerch.

**Methodology:** Aurel Holzschuh, Anita Lerch, Daniel J. Bruzzese.

**Project administration:** Bakar S. Fakih, Safia Mohammed Aliy, Mohamed Haji Ali, Mohamed Ali Ali, Joshua Yukich, Manuel W. Hetzel, Cristian Koepfli.

**Resources:** Safia Mohammed Aliy, Mohamed Haji Ali, Mohamed Ali Ali, Cristian Koepfli.

**Software:** Anita Lerch.

**Supervision:** Aurel Holzschuh, Cristian Koepfli.

**Validation:** Aurel Holzschuh.

**Visualization:** Aurel Holzschuh.

**Writing – original draft:** Aurel Holzschuh.

**Writing – review & editing:** Aurel Holzschuh, Anita Lerch, Bakar S. Fakih, Daniel J. Bruzzese, Joshua Yukich, Manuel W. Hetzel, Cristian Koepfli.

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
