## [Decision Letter · Decision Letter 0]

3 Nov 2023

PGPH-D-23-01739

Real-time genomic surveillance of Plasmodium falciparum using a mobile nanopore sequencing lab: a feasibility study

Dear Dr. Holzschuh,

Thank you for submitting your manuscript to PLOS Global Public Health. After careful consideration, we feel that it has merit but does not fully meet PLOS Global Public Health’s publication criteria as it currently stands. Therefore, we invite you to submit a revised version of the manuscript that addresses the points raised during the review process.

The reviewers both gave overall very positive feedback on the study and have put forward several minor suggestions to improve the clarity and enhance the discussion of the study. It will be particularly helpful (for other readers and potential MinION users) to address the queries and comments concerning:

Query about access to code within the pipeline (reviewer 1)Adding clarity on the marker selection (reviewer 1)Threshold queries (comments 6 and 10 of reviewer 1)Reusing flow cells - is there evidence to back this up for public health/ surveillance purposes? (reviewer 2)Scope for using a high-performance desktop computer for data processing (reviewer 2)

We look forward to receiving your revised manuscript.

Kind regards,

Sarah Auburn

Academic Editor

Journal Requirements:

1. Please include a complete copy of PLOS’ questionnaire on inclusivity in global research in your revised manuscript. Our policy for research in this area aims to improve transparency in the reporting of research performed outside of researchers’ own country or community. The policy applies to researchers who have travelled to a different country to conduct research, research with Indigenous populations or their lands, and research on cultural artefacts. The questionnaire can also be requested at the journal’s discretion for any other submissions, even if these conditions are not met.  Please find more information on the policy and a link to download a blank copy of the questionnaire here: https://journals.plos.org/globalpublichealth/s/best-practices-in-research-reporting. Please upload a completed version of your questionnaire as Supporting Information when you resubmit your manuscript.”

2. Please provide separate figure files in .tif or .eps format.

Additional Editor Comments (if provided):

Reviewers' comments:

Reviewer's Responses to Questions

**Comments to the Author**

1. Does this manuscript meet PLOS Global Public Health’s publication criteria? Is the manuscript technically sound, and do the data support the conclusions? The manuscript must describe methodologically and ethically rigorous research with conclusions that are appropriately drawn based on the data presented.

Reviewer #1: Yes

Reviewer #2: Yes

2. Has the statistical analysis been performed appropriately and rigorously?

Reviewer #1: Yes

Reviewer #2: Yes

3. Have the authors made all data underlying the findings in their manuscript fully available (please refer to the Data Availability Statement at the start of the manuscript PDF file)?

Reviewer #1: Yes

Reviewer #2: Yes

4. Is the manuscript presented in an intelligible fashion and written in standard English?

Reviewer #1: Yes

Reviewer #2: Yes

5. Review Comments to the Author

Reviewer #1: The authors developed and validated a two-multiplexed amplicon P. falciparum sequencing assay using the MinION that incorporated 6 microhaplotypes and ten drug resistance markers. This assay was then tested in a mobile lab setting in Zanzibar.

Overall, this is a valuable pilot study bringing a proof-of-concept of using the MinION (and a mobile lab) to generate valuable data in country also highlighting potential challenges of this approach.

Please see my comments below:

1) Data availability – currently pending so not (yet) accessible?

2) Title: this is set as real-time genomic surveillance and although the technology does show fast results, I would argue these not to be real-time (the paper states “actionable data within a few days”). The authors could amend their title or maybe clarify the real-time aspect of it further.

3) The authors state they developed a pipeline that allows genotyping of polyclonal infections. Technically I did not see a pipeline (maybe one could call it a framework?). I was not able to access the code (related to point 1) – but depending on how the code is shared and commented on, the author could refer to the code repository in the method section of this paper. This would remove the need to add code snippets and keep the method section clean (i.e. just note which software was used and then refer to code).

4) Could the authors clarify as to why they have limited the sequencing to 16 markers and why they chose the markers they did? Is there an option to extend? If yes - that would be valuable to highlight. Was it two multiplexed to evaluate the barcode cross-talk? Is there the possibility to put it into one multiplex?

5) Differences in chemistry – as ONT states that their new chemistry comes with better quality (and more data), it might be worth clarifying the differences in the data analysis (i.e. separating outcomes in regards to median quality, run time, read amount etc between old and new chemistry). Since the old chemistry will no longer be supported or sold by ONT, it would be interesting to showcase the new chemistry results more (are they actually much better?).

6) the authors have calculated a cut-off of >65 reads per haplotype. If I understood it correctly, they used all samples in one pot to calculate this. I wonder if it would have been more suitable to create a measurement that was relative to the depth of the sample. In our experience the sequencing tends to be uneven between the samples so a hard cut-off could lead to a loss of valuable data in those samples with a lower read amount.

7) Figure 1: both Unguja and Pemba are in a square – maybe add the names to the picture or clarify the border in the legend.

8) Figure S3: the y-axis labelling is slightly confusing. The axis is showing the nr of reads not the log10 nr of reads. I think it is clearly visible that it was plotted using a log10 scale so could possibly just remove from label?

9) Figure 5 – see point 8– misleading axis label

10) general note: the authors are losing a lot of data during their Q15 filter step (over 50%), it might be worth exploring a slightly lower threshold. Nanopore recommends 7, though most I know filter at around 12.

Reviewer #2: Holzschuh et al provide an excellent study addressing the need for field-deployable sequencing assays for ongoing malaria surveillance with the goal to achieve malaria elimination. Without appropriate tools to track malaria transmission and drug resistance, the battle to eliminate malaria will continue. Holzschuh et al choose nanopore technology to address the lack of sequencing throughout Zanzibar. Nanopore is an appropriate sequencing method for this purpose by providing intact long reads, faster sequencing turn-around time but also a cost-effective alternative to short read Illumina sequencing platforms. The small, portable nature of Nanopore, MinION makes it an appealing technology to use for parasite surveillance. This was a proof-of-concept, pilot study to determine whether it would be possible to identify known drug resistant haplotypes quickly and easily. Table 5 clearly identifies the challenges associated around in-country sequencing to which many have struggled with previously including similar studies that have been conducted in Papua New Guinea (PNG). The authors have correctly identified key problems and attempted to identify solutions that could be implemented which seem feasible. Overall, capacity building/strengthening is imperative to the success of genomic surveillance in-country. Similar concepts are being established in other malaria endemic countries. This is a great initiative with huge potential and vital to the success of malaria elimination as drug resistance is increasing.

Consistent with other studies, Holzschuh et al also had difficulty with roughly half of DBS and RDTs collected in the Zanzibar pilot study had low parasite density and therefore were not sequenced. Would be great for teams across the world to come together and determine how to sequence these low parasite density samples as it would be interesting to see what’s happening at these low levels. Something that could strengthen the paper is to discuss ideas on how to increase the number of samples of low parasite density getting to sequencing?

Discussion was had about the issues around basecalling and haplotype calling. These informatic steps require significant computing capacity. Unsurprisingly authors were faced with poor internet for transferring the data. Should the authors wish to continue malaria surveillance in Zanzibar I believe their statement around a powerful GPU computer would be an appropriate investment. A laptop would have some computing restrictions and not be the most appropriate, although convenient would not save time and result it storage limitations or restricted GPU capacity. However, setting up a high range desktop computer in the ZAMEP laboratory and using hard drives to transfer data would be the best option. For example, you could get an Intel Core-i9 13900KS processor, RTX4090 graphics, 128GB DDR5 RAM and over 25TB of storage for around $5000-$8000. This would be an exceptional investment for the lab. You could collect your samples from various sites around Zanzibar, use the mobile lab for processing samples, sequence with MinION and put data output straight onto the computer to analyse. This would save time and most likely not rely on internet connection or the need for a HPC system. The caveat here would be that someone needs to understand the bioinformatics component. This section of the discussion could be improved with the above information or equivalent.

Great to see the team thinking ahead. There a several groups currently working on a graphical user interface tool to make analysis quicker and easier for these in-field studies and surveillance of malaria. Regardless of what app is developed, the bottleneck will always be raw sequence data processing but will be beneficial for quick analysis and visual interpretation of a parasite population or drug resistant mutations.

The authors have highlighted the costs of using Nanopore technology. Providing there were going to be a lot of parasite surveillance, I believe you could bulk order flow cells and reduce the cost to $600 per flow cell. It would also be really good to have explored the idea of reusing flow cells. Washing the flow cell does not guarantee the previous template has been removed. Has there been a feasibility study conducted on whether previous sequence samples are contaminating the new sequencing run? Would this then change the results and give false positives ect?

Minor changes:

1. Line 225: would be good in the discussion to add a couple of points on re-using flow cells. Typically for most accurate results, flow cells aren’t reused. Would be good to make a statement or direct to publication that has tested the success of re-used flow cells compared to unused.

2. In the methods: it would improve the clarity of the paper to include a couple of really clear statements identifying where data analysis was conducted. It is my understanding that the validation work was conducted at the University of Notre Dame and the pilot study conducted in ZAMEP lab. In the discussion it is discussed issue around transferring data to HPCs. In the methods it should be clearly stated that the mk1c was not used for basecalling but instead a HPC was used and where this is.

3. Ensure consistency: In line 230 “version 6.1.7” is stated but in line 215 “v22.05.8” is used. I would change “v” to “version” and keep consistent throughout.

4. Line 670: Would be great to specify other disease control and elimination programs. Give an example. You could survey soil-transmitted helminths for example as well as malaria should the capacity be built in country.

5. Line 621: you could add my comments earlier about investment in a high-performance desktop computer instead of focusing on a laptop. If one of these desktop computers were used, you would actually decrease computing time compared to the MinION device. “Setting up a high range desktop computer in the ZAMEP laboratory and using hard drives to transfer data would be the best option. For example, you could get an Intel Core-i9 13900KS processor, RTX4090 graphics, 128GB DDR5 RAM and over 25TB of storage for around $5000-$8000.”

Great study!

6. PLOS authors have the option to publish the peer review history of their article (what does this mean?). If published, this will include your full peer review and any attached files.

**Do you want your identity to be public for this peer review?** For information about this choice, including consent withdrawal, please see our Privacy Policy.

Reviewer #1: No

Reviewer #2: No

---

## [Editor Report · Decision Letter 1]

11 Jan 2024

Using a mobile nanopore sequencing lab for end-to-end genomic surveillance of Plasmodium falciparum: a feasibility study

PGPH-D-23-01739R1

Dear Mr. Holzschuh,

We are pleased to inform you that your manuscript 'Using a mobile nanopore sequencing lab for end-to-end genomic surveillance of Plasmodium falciparum: a feasibility study' has been provisionally accepted for publication in PLOS Global Public Health.

Best regards,

Sarah Auburn

Academic Editor